# Change in Caffeine Consumption after Pandemic (CCAP-Study) among University Students: A Cross-Sectional Study from Italy

**DOI:** 10.3390/nu16081131

**Published:** 2024-04-11

**Authors:** Giuseppe Di Martino, Pamela Di Giovanni, Federica Vaccaro, Fabrizio Cedrone, Edoardo Trebbi, Livia Tognaccini, Ferdinando Romano, Tommaso Staniscia

**Affiliations:** 1Department of Medicine and Ageing Sciences, “G. d’Annunzio” University of Chieti-Pescara, 66100 Chieti, Italy; pamela.digiovanni@unich.it (P.D.G.); federicavaccaro2@gmail.com (F.V.); tommaso.staniscia@unich.it (T.S.); 2Unit of Hygiene, Epidemiology and Public Health, Local Health Authority of Pescara, 65100 Pescara, Italy; 3Hospital Management, Local Health Authority of Pescara, 65100 Pescara, Italy; cedronefab@gmail.com; 4Department of Public Health and Infectious Diseases, “La Sapienza” University of Rome, 00100 Rome, Italy; edoardo.trebbi@uniroma1.it (E.T.); livia.tognaccini@uniroma1.it (L.T.); ferdinando.romano@uniroma1.it (F.R.)

**Keywords:** caffeine, students, public health, COVID-19, pandemic, Italy

## Abstract

(1) Background: The lockdown, imposed to limit the pandemic evolution, impacted the social habits and lifestyle of inhabitants also from many countries other than Italy. Caffeine consumption could be a useful option to improve mood, as the lockdown strongly affected mental health, particularly increasing anxiety and sleep disorders. (2) Methods: It was a cross-sectional study aimed to evaluate the change in caffeine consumption after the pandemic period in a sample of Italian University students. It investigated coffee and products containing caffeine consumption, the anxiety level by State Trait Anxiety Inventory (STAI) and sleep quality with the Insomnia Severity Index (ISS). (3) Results: A total of 404 participants were enrolled in the study. During the lockdown period, 23% of subjects consumed more coffee compared to the pre-pandemic period. Daily coffee consumption also changed after the lockdown period compared to the pre-pandemic period, with 36.1% of subjects that increased their consumption. Comparing other caffeine-based products, energizing supplement consumption showed a greater increase after the lockdown period than energy drinks. Regarding anxiety, subjects who reported increased coffee consumption reported higher STAI scores and ISS. (4) Conclusions: About a quarter of university students increased their consumption of caffeine-based products after the pandemic, reporting higher levels of anxiety and poor sleep quality.

## 1. Introduction

At the beginning of the year 2020, the Coronavirus disease 2019 (COVID-19) rapidly spread around the globe [1] and it was classified as a global pandemic by the World Health Organization (WHO) [2]. Due to the severity of the acute respiratory syndrome (ARDS), which represents a severe complication of COVID-19 [3], along with the significantly rapid spreading of the virus, governments of countries around the world decided to implement non-pharmacological protective measures in order to effectively decrease the virus transmission [4]. From the beginning of the pandemic, Italy has been one of the most affected countries, so the Italian government declared a national lockdown on 9 March 2020, in order to mitigate the rising infection rate [5]. Wearing masks, isolation and social distancing have heavily affected people’s daily lives almost worldwide, limiting daily routine, particularly in environments such as schools, universities and workplaces [4]. At the same time, lockdown strongly influenced mental health, frequently causing an increase in depressive and anxiety symptoms [6]. The lockdown, imposed to limit the pandemic evolution, thereafter impacted the social habits and lifestyle of inhabitants also from many countries other than Italy [7,8,9].

Drinking coffee and/or other caffeine-containing substances can be useful to improve mood and attention and on the other hand, to reduce stress [10,11,12]. These positive effects could be used by people to face lockdown-related stress and improve mood [11]. So, lockdown could play a role in changing caffeine consumption during the period of isolation and it can be considered as an important proxy of mental health [12]. In fact, recent studies suggested that increased coffee consumption can be considered a way to cope with stress and boredom [13]. In parallel with coffee, energy drink consumption also increased during the pandemic [14]. Energy drinks contain a mixture of various substances, such as amino acids and vitamins, while all contain a high amount of caffeine. Most energy drinks contain approximately between 80 and 350 mg of caffeine per serving compared to 100 mg of caffeine contained in a single coffee cup [14]. Excessive caffeine consumption can strongly impact physical and mental health: sleep disorders and worsening in sleep quality are frequently reported [15], accompanied by other physical symptoms such as headaches, nausea, gastrointestinal disorders and increased heart rate [16,17]. Mental symptoms can also be associated with the consumption of energy drinks mixed with alcohol, which promotes anxiety and irritability [18,19]. Despite the caffeine consumption potentially being be a useful option to improve mood, the lockdown strongly affected mental health, particularly increasing anxiety and sleep disorders, especially among university students who attended lessons and exams via the web, requiring more attention compared to habitual university attendance [13]. A poor number of studies have been performed in order to evaluate the possible change in caffeine consumption among university students, whose behaviors can lead both to worse physical and mental outcomes. Evaluating harmful behaviors among students can be important to develop preventive strategies and improve students’ health.

The “Caffeine Consumption After Pandemic—(CCAP Study)” was designed to evaluate the change in caffeine consumption in a sample of Italian University students and its association with different mental health outcomes; in particular, anxiety and sleep disorders. Despite some studies investigating the change in caffeine consumption during the pandemic, no studies have evaluated its association with mental health.

## 2. Materials and Methods

The CCAP study is a cross-sectional study aimed to evaluate the change in caffeine consumption after the pandemic period in a sample of Italian university students from the Italian Southern Region.

### 2.1. Study Design

In December 2022, the authors started designing the study. The questionnaire was developed after a literary review, particularly evaluating the most important mental health domains that need to be investigated in association with caffeine consumption. After the identification of mental health domains, appropriate tools were searched. In particular, this study needed reliable tools already validated in the Italian version and that can be self-administrated. In addition, sociodemographic factors were included in order to evaluate their association with study outcomes. Selected tools and socio-demographic items were then merged into a single questionnaire reported on Google Forms (©2022 Google, Mountain View, CA, USA). All students attending the “G. d’Annunzio” University of Chieti-Pescara were eligible to participate. The questionnaire link was shared via social media or directly promoted to students during lessons. In order to improve participation, enrolled students were also incentivized to promote the survey among their colleagues. Participation was voluntary. The enrollment procedure started in February 2023 and the last participant was enrolled in September 2023.

### 2.2. Sample Size Estimation

The minimum required sample size was calculated according to the subsequent formula:$$\mathrm{Sample}\;\mathrm{size}=\frac{\frac{z^{2}\times p\;(1-p)}{e^{2}}}{\frac{1+(z^{2\;}\times p(1-p)}{e^{2}N}}$$

Considering a total of 21,694 students attending the “G. d’Annunzio” University of Chieti-Pescara (academic year 2021/2022), with a 95% confidence interval and a 5% margin error, at least 378 students should be enrolled.

### 2.3. Informed Consent and Ethical Approval

The cover letter of the questionnaire briefly reported the aim of the study, informing participants of the scientific purposes of the study. Google Forms was used only to spread the questionnaire and to collect data. All the collected data were then downloaded and stored in a professional computer and protected by a password. Participants filled out the informed consent prior to the survey participation.

The study does not require the ethical committee’s approval because the questionnaire data were totally anonymous, making it impossible to identify and harm any participant. Moreover, substances, drugs nor medical devices were prescribed/administered. As a result, results were examined and reported as aggregate, in accordance with Italian and European regulations on the management of personal data (DL 101/2018 and GDPR 2016/679) [20,21].

### 2.4. Questionnaire

The final version of the survey was composed of five main sections. The first section reported sociodemographic characteristics of enrolled subjects, investigating demographic characteristics, region of provenience and degree course attended. The second section reported information about COVID-19 vaccination and COVID-19 disease history. Participants were asked to report if they were vaccinated against COVID-19, the number of jabs received, history of COVID-19 infection and time from the last COVID-19 infection. The third section investigated coffee and consumption of products containing caffeine. In particular, the consumption of coffee, energy drinks and other products containing caffeine was investigated, exploring the relative frequency of consumption before, during and after the pandemic periods. In addition, the reason for consumption or the occurrence of adverse effects due to the consumption was also asked. The fourth section investigated the anxiety level of participants with the State Trait Anxiety Inventory (STAI) [22]. STAI explores the anxiety state and the anxiety trait with twenty statements that evaluate how the respondent feels “right now, at this moment”, on a 4-point scale: (1) not at all, (2) somewhat, (3) moderately, (4) very much so. Higher scores represent higher levels of anxiety. A cutoff score of 40 is commonly used to define probable clinical levels of anxiety. The Cronbach’s alpha in the Italian validation study was 0.64. The final section explored the severity of insomnia of each participant. It was assessed with the Italian version of the Insomnia Severity Index (ISI) [23]. This is a validated scale aimed at evaluating the severity of night-time and daytime components of insomnia. It consisted of seven questions whose results were summarized to obtain a total score. From a score ranging from 0 to 7, insomnia is not clinically significant, 8–14 was referred to subthreshold insomnia, 15–21 represented a moderate severity of insomnia and 22–28 was referred to as severe clinical insomnia. The Cronbach’s alpha of this questionnaire in its validation study was 0.76.

### 2.5. Statistical Analysis

Data were presented as frequency and percentage for categorical variables and mean and standard deviation or median and interquartile range (IQR) for continuous variables, according to their distribution. Kolmogorov–Smirnoff’s test was performed to test the normal distribution of each continuous variable. Categorical variables were compared with Pearson’s chi-squared test or Fisher’s exact test as appropriate. Paired qualitative variables were analyzed with the McNemar test. Continuous variables were compared with the Student’s *t*-test of the Mann–Whitney U test as appropriate. Multivariable logistic regression models were performed to evaluate factors associated with the change in caffeine consumption after the pandemic period. Spearman correlation was performed to evaluate the correlation between ISS and STAI.

A *p*-value < 0.05 was considered statistically significant. The statistical analyses were carried out with STATA v.14 (StataCorp LLC, College Station, TX, USA).

### 2.6. Ethical Statement

This study does not require ethical committee approval because the questionnaire data are totally anonymous, making it impossible to identify and harm any respondent. Moreover, drugs nor medical devices were prescribed/administered. As a result, the responses are collectively examined while taking into account Italian and European regulations governing the management of personal data [21,24,25].

The cover letter of the questionnaire informed participants that the data were used only for scientific purposes, archived for a maximum of five years and were accessible only by the members of the study group. All collected data were stored in a computer and protected by a password, accessible only to study group participants.

It was possible to complete the questionnaire only after the participants declared that they understood the methods and purpose of the study by clicking on the “I give the consent” option in reference to the processing of personal data. Participants who disagreed ended the survey without responding to any other questions.

## 3. Results

A total of 404 participants were enrolled in the study and completed the survey successfully. The majority were women (302, 74.8%), while the number of males was three times lower (102, 25.2%). The mean age was 22.1 ± 3.9 years. Nearly all of total came from the southern Italian region (377, 93.3%) and only three participants were foreigners (0.7%). The most represented degree course attended was pharmacy (99, 24.5%), followed by nursing (71, 17.6%) and midwifery (49, 12.1%). Among other degree courses (150, 37.1%), 110 of 150 participants were related to other healthcare sciences (laboratory technicians, dietary sciences, radiology technicians, healthcare assistants and dentistry). Only 40 participants (9.9%) were related to non-health-related courses such as economy and law. The majority of enrolled participants reported being vaccinated against COVID-19 (403 of 404, 99.8%) and reported a previous COVID-19 positivity (289, 71.5%). Among infected patients, 242 of them (83.7%) reported positivity more than 6 months ago. Only 16 patients (5.6%) reported being infected during the last three months. The median STAI-Y score was 98 (IQR 79–110) and the median ISS score was 11 (IQR 7–16). Regarding STAI-Y, the median score of Anxiety state score was 50 (IQR 40–57), while the median Anxiety trait score was 48 (IQR 39–56). Baseline characteristics of enrolled patients are reported in Table 1.

Regarding coffee consumption, 137 (33.9%) participants reported consuming 2/3 of coffee per day and 91 (22.6%) reported consuming no daily coffee. During the lockdown period, the no-consumers group was less represented compared to the actual period (127 participants, 31.4%). In the period before the lockdown, non-consumers represented 134 (33.2%). The results of coffee consumption are reported in Table 2.

Regarding energy drink consumption, non-users represent the majority of enrolled participants (291, 72%), in parallel with energizing food supplement consumption (256, 63.3%). Energy drink non-users represented a great part of subjects both in the lockdown period and in the pre-pandemic period, respectively, 323 (80%) and 308 (76.2%). It was the same situation among energizing supplements, with non-users representing, respectively, 72.8% of subjects (*n* = 294) in the lockdown period and 69.3% (*n* = 280) in the pre-pandemic period. The frequency of consumption for energy drinks and supplements is reported in Table 3.

The main reason for energy drink consumption was “To be more concentrated during study” (38, 31.1%), while the main reason for energizing supplement consumption was “to obtain more energy” (49, 30.4%), as reported in Appendix A. Few subjects reported adverse events after energy drink or supplement consumption (64 subjects), of which the majority reported experiencing tachycardia (33, 51.5%).

During the lockdown period, 93 subjects (23%) consumed more coffee compared to the pre-pandemic period (McNemar test *p* < 0.001). Daily coffee consumption also changed after the lockdown period compared to the pre-pandemic period, with 146 subjects (36.1%) that increased their consumption (McNemar test *p* < 0.001). Comparing other caffeine-based products, energizing supplement consumption showed a greater increase after the lockdown period than energy drinks (67, 16.6% vs. 37, 9.2%, respectively), as reported in Table 4.

Among factors associated with the increase in coffee consumption, age and gender were significantly associated. In particular, the female gender was positively associated with coffee consumption increase (OR 1.72, 95%CI 1.01–2.95, *p* = 0.046) while age was negatively associated (OR 0.79, 95%CI 0.69–0.89, *p* < 0.001). Smoking habits and region of residency did not result as significant associated factors.

Regarding anxiety, subjects who reported an increased coffee consumption reported higher STAI scores. In particular, the median score was 102 (83.8–116.0) among patients with increased consumption, compared to the median score of 95 (78–108.3) among others (*p* = 0.040), as reported in Figure 1.

Also, subjects that reported an increased consumption of energy supplements, reported higher anxiety, with a median score of 107 (79–124) versus a median score of 98 (79.5–109) (*p* = 0.023), as reported in Figure 2.

No differences in STAI were found between ED consumption patterns. In particular, subjects that increased ED consumption reported a median STAI of 100 (IQR 72.5–134.0), compared to a median STAI of 98 (IQR 80.0–110.0) among participants whose ED consumption was not increased (*p* = 0.411).

Regarding the evaluation of insomnia, 132 subjects (32.7%) reported a clinical insomnia pattern. Subjects with clinical insomnia reported significantly higher scores in STAI compared to non-insomnia subjects, respectively, 110 (100–130) and 87 (74–103) (*p* < 0.001). ISS and STAI scores were also positively correlated (Rho: 0.527, *p* < 0.001), as shown in Figure 3.

Subjects that increased caffeine consumption after the pandemic reported higher ISS values compared to other subjects, with a median score of 12 (7–18) versus 11 (7–15) (*p* = 0.014). Significantly higher ISS results were reported also for subjects with increased energizing supplement consumption [15 (9–10) vs. 11 (7–15); *p* < 0.001] and increased energy drink consumption [16 (9–21) vs. 11 (7–15); *p* = 0.004].

## 4. Discussion

The COVID-19 pandemic heavily impacted people’s mental health, causing an increase in mental disorders, particularly in Western countries [24]. There is no doubt that healthcare workers were the most affected persons by mental illness during the pandemic due to the heavy direct activities in assisting COVID-19 patients. On the other hand, people forced to stay home due to national lockdown were also exposed to a higher risk of mental instability. Smart working and obligation to stay home are the main factors associated with the decrease in social contact and physical activity. Students are one of the most vulnerable social groups, possibly strongly impacted by the pandemic. Many students moved in with their parents or other relatives when the pandemic prolonged, and lessons were not performed in person and social contacts were not allowed [14]. Lack of live teaching and class time increased loneliness and isolation. These situations also impacted the quality of learning, causing a decrease in attention and knowledge. The lower quality of learning, in addition to social distancing, surely impacted students’ well-being, increasing anxiety and sleep disorders. Drinking coffee is one of the easiest ways used by students to cope and improve attention and study productivity [25], despite the effects of caffeine on concentration and learning ability being unclear [25].

Initially, we hypothesize that caffeine consumption among Italian university students was not significantly increased, according to recent literature [14]. Our results showed a change in consumption patterns both for coffee and other caffeine-based products. The reasons for the maintenance of higher coffee consumption after the pandemic are unknown. We hypothesize that a worsened mental health condition can lead to less healthy behavior. On the other hand, mental well-being, which was strongly impacted by the pandemic, could be associated with caffeine consumption.

Regarding mental health, university students are a sensitive group that need social interactions that were lacking during the pandemic. Anxiety and depression increased during the pandemic due to isolation from friends and relatives. Furthermore, the stress that accompanies the uncertainty about the future, economic worries, and delays in academic activity due to the pandemic, can also be considered risk factors for developing anxiety. This point confirms previous literature, defining university students as a risk group for mental health distress after the pandemic [21]. It will be important to focus on mental health among students in order to prevent future mental illness and avoidable hospitalizations [26]. These results are also consistent with the OECD report, which indicates a substantial decrease in mental health among young subjects [27]. The data were confirmed also in a recent systematic review [28].

Also, sleep quality was influenced by the pandemic period. In fact, our study demonstrated how a great part of enrolled subjects reported clinically relevant insomnia. Several causes can lead to poor sleep quality. Other than depression linked to isolation, COVID-19 positivity is a known risk factor for insomnia [29]. In addition, anxiety and sleep quality are directly correlated, confirming that anxiety is a known risk factor for poor sleep quality, as reported in several previous studies [30,31,32]. Also, caffeine assumption can influence sleep quality. Despite studies being conducted during the pandemic that did not show evidence of how dietary changes affect sleep quality, caffeine-containing products influence the awake state. The increase in coffee and energy drink consumption can lead to poor sleep quality. In fact, it is known that the timing of the assumption of coffee and energy drinks is sensible, as a great majority of the students consume them before 5 pm. The recommendation regarding the consumption of caffeinated products is to consume them early during the day and preferably not after 3 pm, as the half-life of caffeine is around 4–6 h [14,33].

It is also important to highlight that harmful caffeine consumption can lead to other health risks than anxiety and insomnia [34,35,36]. Some studies showed that caffeine has a negative impact on the nervous system [37], liver [38] and cardiovascular system [39]. Evaluating caffeine consumption among younger subjects, such as university students, could be important to develop public health preventive strategies.

An additional observation should be performed on energy drinks, which are often consumed in combination with alcohol. So, in addition to possible negative direct effects of harmful energy drink consumption on cardiovascular [40], neurological [41] and gastrointestinal systems [42], its impact has to be considered also in association with possible harmful alcohol use.

### Strengths and Limitations

The present study represents one of the first studies conducted in Italy about the change in caffeine-based product consumption after the pandemic among university students. The evaluation of both coffee and other caffeine-based products is a novelty in recent literature. In addition, no previous studies evaluated the association between caffeine consumption and mental health. Anxiety and insomnia were correlated with caffeine consumption using internationally validated scores, making this work replicable in other countries. The sample of the study was homogeneous and typically represented university students in Italy. Among limitations, it is important to highlight that this is a cross-sectional design which is not allowed to directly demonstrate causality between caffeine exposure and mental health. In addition, the previous consumption of caffeine-based products could be wrongly reported due to a recall bias. Further studies with a different design can help to better evaluate the association between caffeine consumption and mental health.

## 5. Conclusions

The present study highlighted that about a quarter of enrolled subjects increased the consumption of caffeine-based products after the pandemic in a sample of Italian students. As reported in this study, it can influence the mental health of students leading also to poor sleep quality. Further studies are needed to evaluate factors associated with this increased consumption in order to develop preventive strategies aimed at limiting the harmful use of caffeine-based products. The association between caffeine and mental health should also be considered an important topic for public health.

## Figures and Tables

**Figure 1 nutrients-16-01131-f001:**
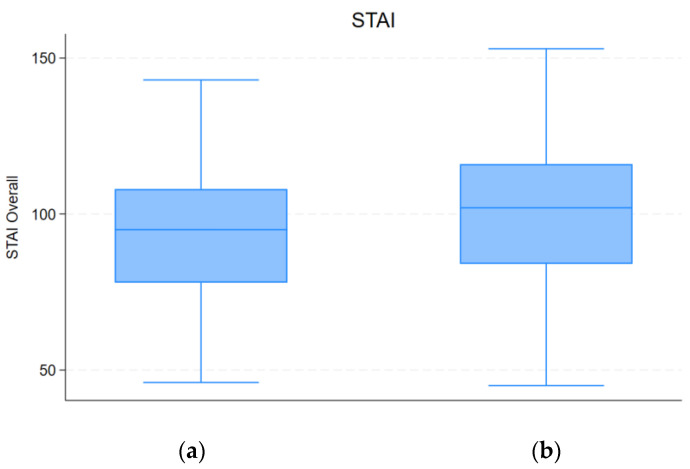
Differences in STAI between participants who increased caffeine coffee consumption (**b**) and participants whose consumption remained equal or lesser (**a**).

**Figure 2 nutrients-16-01131-f002:**
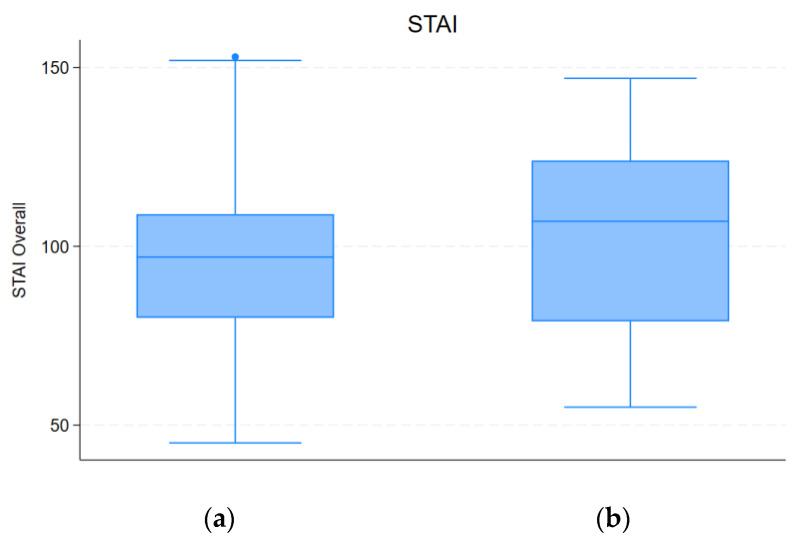
Differences in STAI between participants who increased energy supplement consumption (**b**) and participants whose consumption remained equal or lesser (**a**).

**Figure 3 nutrients-16-01131-f003:**
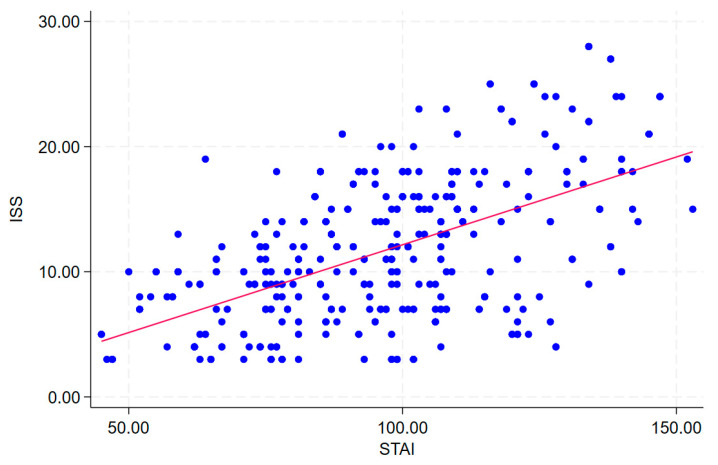
Correlation between STAI and ISS.

**Table 1 nutrients-16-01131-t001:** Baseline characteristics.

Variables	N	%
Age *mean* ± *SD*	22.1 ± 3.9	
Gender		
Female	302	74.8
Male	102	25.2
Region		
Northern	1	0.2
Centre	23	5.7
Southern	377	93.3
Foreigner	3	0.7
Degree course		
Pharmacy	99	24.5
Nursing	71	17.6
Midwifery	49	12.1
Medicine	35	8.7
Others	150	37.1
Vaccinated against COVID-19	403	99.8
Previous COVID-19 diseases	289	71.5
Last COVID-19 positivity		
>6 months	242	83.7
between 3 and 6 months	31	10.7
<3 months	16	5.6
STAI-Y Overall *median (IQR)*	98	(79–110)
Anxiety state *median (IQR)*	50	(40–57)
Anxiety trait *median (IQR)*	48	(39–56)
ISS *median (IQR)*	11	(7–16)

SD: standard deviation; IQR: interquartile range.

**Table 2 nutrients-16-01131-t002:** Daily coffee consumption.

Frequency	N	%
Actual daily coffee consumption		
None	91	22.6
1	112	22.7
2–3	137	33.9
4 or more	64	15.8
Lockdown daily coffee consumption		
None	127	31.4
1	110	27.2
2–3	116	28.8
4 or more	51	12.6
Before lockdown daily coffee consumption		
None	134	33.2
1	113	28
2–3	127	31.4
4 or more	30	7.4

**Table 3 nutrients-16-01131-t003:** Energy drinks and energizing food supplement consumption frequency.

Frequency	N	%	Frequency	N	%
**Actual weekly energy drink consumption**			**Actual weekly energizing food supplement consumption**		
Never	291	72	Never	256	63.3
Less than once/week	73	18.1	Less than once/week	37	9.2
1–3 times/week	29	7.1	1–3 times/week	54	13.4
More than 3 times/week	11	2.8	More than 3 times/week	57	14.1
**Lockdown weekly energy drink consumption**			**Lockdown weekly energizing food supplement consumption**		
Never	323	80.0	Never	294	72.8
Less than once/week	50	12.4	Less than once/week	45	11.1
1–3 times/week	26	6.4	1–3 times/week	30	7.4
More than 3 times/week	5	1.2	More than 3 times/week	35	8.7
**Before lockdown weekly energy drink consumption**			**Before lockdown weekly energizing food supplement consumption**		
Never	308	76.2	Never	280	69.3
Less than once/week	64	15.8	Less than once/week	50	12.4
1–3 times/week	26	6.5	1–3 times/week	44	10.9
More than 3 times/week	6	1.5	More than 3 times/week	30	7.4

**Table 4 nutrients-16-01131-t004:** Change in caffeine-based product consumption compared to pre-pandemic period.

Consumption Frequency	Coffee n (%)	Energy Drinks n (%)	Energizing Supplements n (%)
**During lockdown**			
Greater	93 (23.0)	16 (4.0)	18 (4.5)
Equal	257 (63.6)	359 (88.9)	348 (86.1)
Less	54 (13.4)	29 (7.2)	38 (9.4)
*p*-value *	<0.001	0.059	0.004
**After lockdown**			
Greater	146 (36.1)	37 (9.2)	67 (16.6)
Equal	216 (53.5)	350 (86.6)	322 (79.7)
Less	42 (10.4)	17 (4.2)	15 (3.7)
*p*-value *	<0.001	0.041	0.001

* McNemar test for paired data comparing frequency consumption between lockdown/after lockdown period with pre-pandemic period.

## Data Availability

Data was available after reasonable request due to privacy restriction.

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
