# Peer review of "Change in Caffeine Consumption after Pandemic (CCAP-Study) among University Students: A Cross-Sectional Study from Italy"

_nutrients, 2024, doi:10.3390/nu16081131_

Round 1

Reviewer 1 Report

Comments and Suggestions for Authors

The summary is well-written and contains all the elements that are prescribed.

Methods

The work is scientifically sound and the experimental part is suitable for the hypothesis, and the results of the work are reproducible based on the information provided in the methods section.

Results

The figures and tables show the data correctly and are very clear.

Table 1 shows that 37.1% of the students belonged to the "Other" group. It would be good to indicate whether these are health science students like the others (pharmacy, nursing, midwifery and medicine) or whether they are non-health studies. This is important information in view of the assumption that students who study health sciences have more knowledge about nutrition and healthy lifestyles.

Along with Table 3, it would be good to mention which energy-boosting supplements they were taking.

Although the authors talked about the conclusions of their research at the end of the Discussion chapter, it would be good if the conclusion was separated as a separate chapter.

Author Response

Reviewer 1:

  • The summary is well-written and contains all the elements that are prescribed.

The work is scientifically sound and the experimental part is suitable for the hypothesis, and the results of the work are reproducible based on the information provided in the methods section.

Reply: We thank the Reviewer for the appreciation of our work;

  • Table 1 shows that 37.1% of the students belonged to the "Other" group. It would be good to indicate whether these are health science students like the others (pharmacy, nursing, midwifery and medicine) or whether they are non-health studies. This is important information in view of the assumption that students who study health sciences have more knowledge about nutrition and healthy lifestyles.

Reply: Thanks for the important comment. The great part of students were referred to health sciences courses. Less than 10% of them were referred to other courses such as economy and law. We know that health sciences students could report a better knowledge about health related behaviours, but the aim of this study was not to analyse factors associated to caffeine consumption, but to evaluate the behaviour independently to subject knowledge. We added the requested information in result section (lines 184-188).

  • Along with Table 3, it would be good to mention which energy-boosting supplements they were taking.

Reply: As reported in methods section (line 130), only caffeine-based supplements were considered.

  • Although the authors talked about the conclusions of their research at the end of the Discussion chapter, it would be good if the conclusion was separated as a separate chapter.

Reply: We added the conclusion section (lines 328-333)

Reviewer 2 Report

Comments and Suggestions for Authors

This paper reported the change in caffeine-based product consumption among Italian university students after pandemic. The consumption of coffee and the products was found to be increased by several percentages of the participants. The study can provide some information on the potential inducer for the high levels of anxiety and poor sleep quality of university students. Before consideration for publication, some issues should be conducted.

Specific comments:

1. In the “Abstract” section, the full name of the abbreviation should be provided when it first appeared. Please revise it.

2. In the introduction part, although the background of the COVID-19 period and coffee consumption, the necessity and the procedure of this research were missing. I suggested that the authors should present the importance of their study and the main content of their work in the introduction section.

3. Discussion part should be revised. The obtained data and information after questionnaire survey can offer some perspectives and recommendations for the coffee consumption and the impact on the health of university students. However, the existed descriptions were not enough. More illustrations should be presented to clarify the purpose of this interesting work.

4. Some detailed data can be added in the “4.1 Strength and limitations” section, which will make the presentation more convincing.

5. Please check and unify the format of references in accordance with the guideline of Nutrients.

Author Response

Reviewer 2:

  • In the “Abstract” section, the full name of the abbreviation should be provided when it first appeared. Please revise it.

Reply: Done.

  • In the introduction part, although the background of the COVID-19 period and coffee consumption, the necessity and the procedure of this research were missing. I suggested that the authors should present the importance of their study and the main content of their work in the introduction section.

Reply: We added the aim of our study in Introduction section (Lines 68-70).

  • Discussion part should be revised. The obtained data and information after questionnaire survey can offer some perspectives and recommendations for the coffee consumption and the impact on the health of university students. However, the existed descriptions were not enough. More illustrations should be presented to clarify the purpose of this interesting work.

Reply: The discussion section was improved accordingly to Reviewer comment (lines 320-330 and 346-351).

  • Some detailed data can be added in the “4.1 Strength and limitations” section, which will make the presentation more convincing.

Reply: Section 4.1 was improved.

Round 2

Reviewer 2 Report

Comments and Suggestions for Authors

It can be accepted.